# Comparative Efficacy and Safety of Tirbanibulin for Actinic Keratosis of the Face and Scalp in Europe: A Systematic Review and Network Meta-Analysis of Randomized Controlled Trials

**DOI:** 10.3390/jcm11061654

**Published:** 2022-03-16

**Authors:** Markus V. Heppt, Igor Dykukha, Sara Graziadio, Rafael Salido-Vallejo, Matt Chapman-Rounds, Mary Edwards

**Affiliations:** 1Department of Dermatology, Universitätsklinikum Erlangen, Friedrich-Alexander-University Erlangen-Nürnberg (FAU), Ulmenweg 18, 91054 Erlangen, Germany; 2Comprehensive Cancer Center Erlangen-European Metropolitan Area of Nuremberg (CCC ER-EMN), 91054 Erlangen, Germany; 3Medical Affairs, Almirall Hermal GmbH, Scholtzstrasse 3, 21465 Reinbek, Germany; igor.dykukha@almirall.com; 4York Health Economics Consortium (YHEC), Enterprise House, Innovation Way, University of York, York YO10 5NQ, UK; sara.graziadio@york.ac.uk (S.G.); mary.edwards@york.ac.uk (M.E.); 5Department of Dermatology, University Clinic of Navarra, School of Medicine, University of Navarra, Avda. Pio XII, 36, 31008 Pamplona, Spain; rsalidov@unav.es; 6Quantics Biostatistics, Exchange Tower, 19 Canning Street Fourth Floor, Canning St, Edinburgh EH3 8EG, UK; matt.chapman-rounds@quantics.co.uk

**Keywords:** actinic keratosis, tirbanibulin, topical treatment, face and scalp, efficacy, safety, systematic literature review, network meta-analysis, dermatology

## Abstract

Actinic keratosis (AK) is a chronic skin condition that may progress to cutaneous squamous cell carcinoma. We conducted a systematic review of efficacy and safety for key treatments for AK of the face and scalp, including the novel 5-day tirbanibulin 1% ointment. MEDLINE, PubMed, Embase, Cochrane Library, clinical trial registries and regulatory body websites were searched. The review included 46 studies, of which 35 studies included interventions commonly used in Europe and were sufficiently homogenous to inform a Bayesian network meta-analysis of complete clearance against topical placebo or vehicle. The network meta-analysis revealed the following odds ratios and 95% credible intervals: cryosurgery 13.4 (6.2–30.3); diclofenac 3% 2.9 (1.9–4.3); fluorouracil 0.5% + salicylic acid 7.6 (4.6–13.5); fluorouracil 4% 30.3 (9.1–144.7); fluorouracil 5% 35.0 (10.2–164.4); imiquimod 3.75% 8.5 (3.5–22.4); imiquimod 5% 17.9 (9.1–36.6); ingenol mebutate 0.015% 12.5 (8.1–19.9); photodynamic therapy with aminolevulinic acid 24.1 (10.9–52.8); photodynamic therapy with methyl aminolevulinate 11.7 (6.0–21.9); tirbanibulin 1% 11.1 (6.2–20.9). Four sensitivity analyses, from studies assessing efficacy after one treatment cycle only, for ≤25 cm^2^ treatment area, after 8 weeks post-treatment, and with single placebo/vehicle node confirmed the findings from the base case. Safety outcomes were assessed qualitatively. These results suggest that tirbanibulin 1% offers a novel treatment for AK, with a single short treatment period, favourable safety profile and efficacy, in line with existing topical treatments available in Europe.

## 1. Introduction

Actinic keratosis (AK) is a chronic, recurrent skin condition caused by long-term sun exposure, which leads to skin damage presenting as small, red, rough, scaly lesions [1]. These lesions are often asymptomatic but may be sore or itch [2]. AK is a heterogenous condition in its pathophysiology, clinical manifestation, histologic features and disease course [3]. Data on the prevalence of AK in Europe is still scarce [4,5], although prevalence increases with age, is positively correlated with male gender [6], and is highest in countries with a high proportion of fair-skinned people and high sun exposure [7,8]. According to a World Health Organization (WHO) report in 2006, which contains the most recent data presenting Europe-wide AK prevalence, there were 131,433,084 cases of AK across Europe in 2006 [9], around 18% of the population [10].

AK is the most common precursor for cutaneous squamous cell carcinoma (cSCC) [7,11]. There is a risk that cases of AK of any severity [12] may develop into cSCC if unidentified and/or untreated, with approximately 10% of untreated patients developing cSCC [13]. AK lesions should be treated to avoid both impact on patients’ health-related quality of life (especially in those with severe AK) [8,14] and the costly care and high mortality rates associated with advanced cSCC [15].

Management of AK generally incorporates the use of sun protection as a basic measure [16], while treatment is designed to reduce the total number of lesions on the skin. Once diagnosed, the choice of treatment depends on the clinical grade, site, size of the area affected, the number of lesions, and presence or absence of field cancerization, when large areas of cells in the vicinity of visible lesions are affected by genetic changes [17]. Interventions commonly include field-directed treatments, such as photodynamic therapy (PDT) and topical diclofenac, imiquimod, 5-fluorouracil [18], and lesion-directed treatments such as cryosurgery. Patients with field cancerization are recommended to undergo either field-directed treatment or a combination of field-directed treatment and lesion-directed treatment [19].

Tirbanibulin is a new chemical substance, which, as a topical formulation, has been developed for the treatment of AK. Tirbanibulin as 10 mg/g (1%) ointment is approved by the European Medicines Agency (EMA, 2021) [20] and the Medicines and Healthcare Products Regulatory Agency (MHRA, 2021) for 5-day topical field-directed therapy of non-hyperkeratotic, non-hypertrophic AK (Olsen Grade 1) on the face and scalp in adults [20]. Tirbanibulin prevents the proliferation of atypical keratinocytes, primarily by inhibiting tubulin polymerization and specifically by inducing apoptosis.

We conducted a systematic review of clinically relevant interventions for AK. This manuscript focuses on a subset of treatments available as common practice in Europe. Interventions and specified outcomes relevant to the European perspective were assessed using qualitative analyses and network meta-analysis (NMA) to investigate the comparative efficacy and safety of treatments.

Assessing the comparative efficacy and safety of treatments for AK is complicated by heterogeneity in study designs [21] and interventions, the difficulty of determining the most appropriate timepoint for assessment (particularly for longer term outcomes) [22], and the nature of AK as a chronic heterogeneous condition in which disease course and response to treatment varies between patients [3,23].

Our NMA included data from two phase three randomized clinical trials (RCTs) (NCT03285477 and NCT03285490) of tirbanibulin 1%. To the authors’ knowledge, tirbanibulin 1% ointment has not yet been compared against existing treatments via NMA. Due to differences in availability across markets, country-specific drug approval, and distinct dosages of approved topical drugs, this NMA is European-focused and intended to provide guidance to health technology assessments, primarily in Europe.

## 2. Materials and Methods

### 2.1. Systematic Review Methods

This systematic review was undertaken according to the principles of systematic reviewing embodied in the Cochrane handbook [24], and the protocol was registered on the PROSPERO database [25] (CRD42020194104). A PRISMA checklist is presented in Appendix A. 

RCTs in adult patients with grade I to III AK on the face and/or scalp were eligible for inclusion in the review. RCTs solely in organ transplant patients were ineligible as this subgroup of patients is not representative of the general population. Eleven specified interventions, including cryosurgery, topical treatments, and PDT were eligible. Only treatments and scheduling currently used in Europe are reported in this paper, with the exception of ingenol mebutate 0.015% (Picato). Although ingenol mebutate 0.015% was recently withdrawn from the European market [26], it was included in the qualitative analyses and NMA as it has previously been widely referenced in European health technology evaluations [27,28], and its inclusion increased the stability of the network.

The outcomes of interest to the review were complete and partial clearance, lesion count reduction, recurrence, adverse events (AEs), local skin reactions (LSRs), and discontinuations due to AEs or LSRs. The European perspective analyses focused on the review outcomes deemed to be most clinically relevant: complete clearance, lesion count reduction, number of severe LSRs, and treatment-related discontinuation. Severe LSRs were specifically: severe redness/erythema, severe flaking/scaling/dryness, severe scabbing/crusting, severe erosion/ulceration, severe vesicles, severe swelling/oedema, severe itching/pruritus, and severe weeping/exudate.

For the outcome treatment-related discontinuation, we considered outcomes described as discontinuations due to treatment-emergent AEs (TEAEs), treatment-related AEs (TRAEs), local AEs, or LSRs, to be relevant. Discontinuations described in these ways were deemed to be discontinuations possibly related to the study treatment, rather than reflecting the study methods, baseline characteristics of the study population, or events that were independent of the intervention.

The full eligibility criteria for the review are detailed in Table 1.

A MEDLINE (OvidSP) search strategy (Appendix A) was designed to identify RCTs on the interventions of interest for patients with AK of the face or scalp and was translated appropriately for six further databases (Appendix A). Searches of trial registers, regulatory body websites and systematic review resources were also conducted (Appendix A). No date or language restrictions were applied to the searches.

A single reviewer assessed the search results and removed the obviously irrelevant records, such as those about ineligible diseases or conducted in children. Following this, titles, abstracts and then full texts were screened by double independent reviewers with any disagreements adjudicated by a third reviewer.

Where results for one trial were reported in more than one paper, all related papers were identified and grouped together to ensure that participants in individual trials were only included once. Data extraction and quality assessment (using the Cochrane Risk of Bias tool version 1.0 [24]) were undertaken by a single reviewer with a second reviewer checking all data points. For each outcome, data were extracted at all time points reported. Papers reporting pooled studies were only used where individual study data were not available for either of the studies being pooled.

The Cochrane Risk of Bias tool [24] considers seven criteria. Studies that adequately addressed all seven criteria were judged to be of ‘high quality’. When one or more of the criteria were rated as ‘unclear’ (i.e., insufficient information was reported to assess the criteria) but all the other criteria were well addressed, the study was judged to have an overall ‘unclear’ risk of bias. When one or more of the criteria were not adequately addressed, the study was considered to have ‘serious methodological concerns’.

### 2.2. Feasibility Assessment Methods

Following data extraction, the similarity of the included trials and their suitability for combining in an NMA was qualitatively assessed in accordance with guidance from the Pharmaceutical Benefits Advisory Committee (PBAC) [29]. Trials were compared based on their designs and risk of bias, characteristics of the recruited patient population, interventions included (e.g., doses, frequency and duration of treatment), outcomes reported (including timepoints of assessment and follow-up) and outcome measures used. Any trials deemed excessively methodologically or clinically heterogenous were not eligible for inclusion in any synthesis (see Appendix A).

### 2.3. Statistical Analysis Methods

Data were prioritized in this order: intent-to-treat population, data for the full analysis set, data for the per-protocol population. Outcome data for completers were only used when no other data were available.

Bayesian NMA was applied to the proportion of patients experiencing the outcome of interest. A regression model with a binomial likelihood and a logit link function was used [30]. Relative treatment effects were estimated as log odds ratios (LORs) and were transformed to odds ratios (ORs) for presentation, with 95% credible intervals (CrI) also reported. Model fit was assessed using deviance information criterion.

Both fixed and random effects models were fitted. Due to the heterogeneity of the studies included in the NMA, random effects models were deemed more appropriate and are reported in this manuscript. Networks were plotted as node = treatment, edge = study, and were examined for connectedness.

Non-informative prior distributions were used, with trial-specific baseline and treatment effects assigned Normal (0, 1000) priors. For random effects models, a weakly informative prior distribution was used for the between-study heterogeneity parameter as the number of links in the networks to inform the estimate of this parameter was relatively low. A log-normal (−2.29, 1.582) distribution was used, as suggested by Turner et al. [31], for between-study heterogeneity when analyzing ‘symptoms reflecting continuation/end of condition’ data.

Studies with zero counts for both/all arms did not contribute evidence to the network. This is a consequence of modeling relative treatment effects and is not surmountable by methods such as adding 0.5 to every arm [30]. Therefore, studies with zero counts for both/all arms were excluded from the NMA.

Heterogeneity was quantitatively assessed for all contrasts in each network informed by two or more studies. Pairwise meta-analyses were conducted and quantified with the I^2^ statistic. Inconsistency was assessed using the node-splitting method [32] where feasible within the network geometry.

A Markov chain Monte Carlo (MCMC) method was used to estimate posterior LORs, and conducted in JAGS (version 4.3.0 [33]) and R (version 3.6.1 or later [34]) with the ‘rjags’ package [35]. Pair programming was conducted in Python (version 3.7.1 [36]) using the ‘pystan’ package [37,38].

### 2.4. Timepoints for Outcome Assessment

Given that each intervention of interest has a different recommended (per label) length of treatment and expected time to optimum patient outcomes, to apply a blanket “timepoint of interest” on a per outcome basis would have increased heterogeneity. Instead, the point at which each outcome was assessed was specific to each intervention. The following list contains references to the labeling data used to inform these decisions. Not all formulations/doses were available in all markets at the time of conducting the analyses.

The following timepoints were established prior to the conduct of the NMA:Tirbanibulin 1% (TIRBA1%): 57 days after start of treatment5-fluorouracil 5% (5FU5%) and 5-fluorouracil 4% (5FU4%) [39]: 4 weeks post-treatment5-fluorouracil 0.5% plus salicylic acid 10% (5FU0.5% + SA) [40]: 4 weeks post-treatmentPDT with aminolevulinic acid (ALA_PDT) sensitizer [41]: 12 weeks post-treatmentPDT with methyl aminolevulinate (MAL_PDT) sensitizer [42]: 12 weeks post-treatmentCryosurgery (CRYO): 12 weeks post-treatmentDiclofenac sodium 3% (DICLO3%) [43]: 90 to 120 days following start of treatment (30 to 60 days from end of treatment)Imiquimod 5% (IMQ5%) [44]: 8 to 12 weeks following end of treatmentImiquimod 3.75% (IMQ3.75%) [45]: 8 to 12 weeks following end of treatmentIngenol mebutate 0.015% (IM0.015%) [46]: 57 days after start of treatment

For studies reporting outcomes at multiple timepoints, outcome data were selected to be as close as possible to these timepoints. A sensitivity analysis (see Section 2.5.4) investigated the impact of excluding studies assessing efficacy outcomes at very early timepoints, i.e., less than eight weeks after the end of treatment. 

### 2.5. Sensitivity Analyses of Complete Clearance

#### 2.5.1. Sensitivity Analysis: Single Course Data Only

In the base case NMA, all studies were analyzed together regardless of the number of courses or sessions of treatment assessed. To increase homogeneity across studies, and comparability with trials of TIRBA1%, we ran a sensitivity analysis including only studies assessing one course or session of treatment. The results were compared with those of the base case analysis (including all studies) to identify differences that may have been due to increased heterogeneity in the base case analysis.

#### 2.5.2. Sensivity Analysis: ≤25 cm^2^ Assessment Area Only

Labeling for TIRBA1% [20] suggests treatment across an area of maximum 25 cm^2^, reflecting the design of the included TIRBA1% trials. In order to increase homogeneity of the evidence base with the two studies of TIRBA1%, a sensitivity analysis of the outcome complete clearance was conducted to include only studies in which the skin area assessed is ≤25 cm^2^.

#### 2.5.3. Sensitivity Analysis: Single Placebo Node

The analyses reported in this paper were conducted with two placebo nodes: topical placebo/vehicle (PLAC_TOP) and placebo PDT (PLAC_PDT)(see Supplementary Section 2 for more details). There are different approaches taken in the existing literature. Our methodology is aligned to that of Ezzedine et al. 2021 [47], but Vegter and Tolley 2014 [21] and Gupta 2013 [48] both merged the two placebo nodes to obtain a more compact network. We assessed the impact of these different approaches by conducting a sensitivity analysis of complete clearance in which all placebo/vehicle treatments were pooled into one single node (PLAC).

#### 2.5.4. Sensitivity Analysis: Studies Assessing Outcomes at ≥8 Weeks after Treatment

A sensitivity analysis was conducted to include only studies that assessed efficacy at eight weeks or more after the end of treatment. This analysis was designed to exclude studies assessing efficacy prior to complete epidermal regeneration of the treated area. The regeneration cycle of the epidermis usually ranges from 35–45 days [49], but is subject to environmental differences [50]. In an elderly population, and those with chronically sun-damaged skin, we took eight weeks to be a realistic yet conservative assumption of regeneration time, before which earlier assessment of response could lead to over- or under-estimation of the clearance, as new or residual lesions may be masked by local skin reactions in the treatment area. 

## 3. Results

### 3.1. Results of the Literature Searches and Screening

Searches were conducted between 24 June 2020 and 2 September 2020 and identified 4129 records. Following deduplication, 2712 records were assessed for relevance. A total of 145 documents were excluded at full text screening (see Appendix A). The PRISMA flow diagram is shown in Figure 1, with explanatory details of exclusion reasons in Appendix A.

A total of 46 studies reported in 86 documents were included in the systematic review and are listed in Appendix A. Six additional papers were included but not data extracted (see Appendix A); one of these papers was not published in English and the remaining five papers reported pooled results of studies for which disaggregated data were already available.

Overall, four of the 46 studies (9%) were considered to have a low risk of bias [51,52,53,54], 26 (57%) had an unclear risk of bias [28,55,56,57,58,59,60,61,62,63,64,65,66,67,68,69,70,71,72,73,74,75,76,77,78], and 16 (35%) were considered to have serious methodological concerns [79,80,81,82,83,84,85,86,87,88,89,90,91,92,93,94]. A summary of the risk of bias assessment is shown in Figure 2. 

### 3.2. Results of the Feasibility Assessment

Key characteristics of the 46 studies included in the review are presented in Appendix A. 

Following the qualitative assessment of similarity, 6 of the 46 studies were deemed to be unsuitable for inclusion in the NMA (see Supplementary Section 2.1). One study [94] did not report the number of patients assessed per arm, meaning that insufficient data were available for the NMA. The other five studies [58,76,88,90,93] were unsuitable as the treatment dose, length or schedule assessed differed from both the US and European labels and was insufficiently similar to other included studies assessing the same treatments (see Appendix A for further details).

Due to differences in the US and EU labeling for IMQ5%, studies assessing this intervention were split into those assessing the US (treatment for 16 weeks [96]: IMQ5%_USA) and European (one or two treatment periods of 4 weeks [44]: IMQ5%_EU) schedules. Of the 40 studies remaining, a further five studies [55,56,57,80,83] were not relevant to a European perspective, as they assessed the US posology of IMQ5%. Only studies assessing IMQ5%_EU contributed to the analysis of the Europe perspective. Thirty-five studies (Appendix A) were therefore eligible for inclusion in the analyses. Not all the 35 studies included in the qualitative analyses and NMA reported data for all outcomes; therefore, some interventions are not represented in some analyses.

### 3.3. Qualitative Synthesis: Europe

Following the feasibility assessment, outcomes were summarized through a qualitative synthesis, with the exception of complete clearance for which quantitative synthesis with NMA was feasible. A summary is reported below and full results for outcomes other than complete clearance can be found in Appendix A.

#### 3.3.1. Lesion Count Reduction

No networks assessing this outcome were possible due to insufficient reporting of data by the included studies. Data for the qualitative analysis of lesion count reduction were available for TIRBA1%, MAL_PDT, IM0.015%, DICLO3%, 5FU4%, 5FU5%, 5FU0.5% + SA, ALA_PDT and IMQ3.75%. Definitions of lesion count reduction were not well reported, and it was not always clear whether the included studies reported a mean or median percentage reduction. Further information can be found in Appendix A.

Reported lesion count reductions ranged from 52% (DICLO3% [91]) to 94% (ALA_PDT [52] and 5FU5% [83]) in the active arms and from 14% [57] to 47% [62] in the placebo/vehicle arms. All interventions showed substantial reductions in lesion counts when compared to placebo/vehicle. The comparisons with placebo/vehicle were sparse. 

#### 3.3.2. Discontinuation due to AEs or LSRs

Rates of discontinuation due to TEAEs, TRAEs, local AEs, or LSRs are detailed in Table 2. 

Although data were available on discontinuation due to AEs, it was not possible to perform an NMA due to zero counts in active arms of trials for some interventions, e.g., in both TIRBA1% trials [30] (see “Statistical analysis methods”) which meant that no OR between intervention and placebo/vehicle could be assessed. Data for the qualitative analysis of this outcome were available for TIRBA1%, 5FU0.5% + SA, DICLO3%, ALA_PDT, CRYO, IMQ3.75%, IMQ5%_EU, IM0.015%, MAL_PDT, and 5FU5%. Further information can be found in Appendix A.

Discontinuation rates due to TRAEs, TEAEs, local AEs or LSRs for the active arms ranged from 0% (TIRBA1% [53,54], IMQ5%_EU [28,51]; 5FU5% [28]; MAL_PDT [28]; ALA_PDT [52,85]; CRYO [89]; and IM0.015% [28]) to 12.3% (DICLO3% [71]). Rates for the active arms were generally low, i.e., below 4% for all but two interventions assessed by studies reporting this data. Only studies of DICLO3% [71,75,79] and 5FU0.5% + SA [89] reported rates higher than 4% in the active arms. Rates for placebo/vehicle ranged from 0% [53,54,62] to 3.9% [71]. 

#### 3.3.3. Incidence of Severe LSRs

Data on the incidence of at least one severe LSR following treatment were available for TIRBA1%, DICLO3%, 5FU0.5% + SA, ALA_PDT, IMQ3.75% and IMQ5%_EU. No data on the incidence of specific severe LSRs were available for CRYO, IM0.015%, MAL_PDT, or 5FU (4% or 5%). Data are presented in Table 3 and additional narrative description can be found in Appendix A.

The safety profile of TIRBA1% showed low rates of any single severe LSR (≤11%) with the two studies (NCT03285477 [53] and NCT03285490 [54], respectively) reporting 6% and 11% of patients experiencing severe flaking/dryness, 3% and 10% experiencing redness/erythema, and a maximum of 3% of patients experiencing any of the remaining five severe LSRs reported. This was consistent with the minimal data available for 5FU0.5% + SA and DICLO3% (both assessed by the same study [79]), for which 4.8% and 7% of patients, respectively, experienced severe itching/pruritus. Studies of IMQ5%_EU [61] and IMQ3.75% [59] reported higher rates of severe LSRs, with up to 31% [61] and 25% [59] of patients, respectively, experiencing severe redness/erythema, and 24% [61] and 13.7% [59] experiencing scabbing/crusting. Only one study [74], assessing ALA_PDT, reported that no patients in the intervention or placebo/vehicle arms experienced any severe LSR.

### 3.4. Results of the NMA of Complete Clearance

#### 3.4.1. Base Case Analysis

The base case analysis included all eligible studies relevant to the European perspective, assessing any number of courses of treatment, and reporting the outcome complete clearance. The network diagram and ORs for the base case analysis are presented in Figure 3, with data displayed in Table 4. Details of posology, duration and number of cycles assessed are presented for all studies in Appendix A.

In the base case, all active treatments were associated with higher odds of complete clearance than topical placebo/vehicle (5FU0.5% + SA 7.6 [4.6–13.5]; 5FU4% 30.3 [9.1–144.7]; 5FU5% 35.0 [10.2–164.4]; ALA_PDT 24.1 [10.9–52.8]; CRYO 13.4 [6.2–30.3]; DICLO3% 2.9 [1.9–4.3]; IM0.015% 12.5 [8.1–19.9]; IMQ3.75% 8.5 [3.5–22.4]; IMQ5%_EU 17.9 [9.1–36.6]; MAL_PDT 11.7 [6.0–21.9]; TIRBA1% 11.1 [6.2–20.9]).

5FU5% and 5FU4% had higher ORs than other active treatments, however, wide, overlapping credible intervals were reported. DICLO3% had the lowest OR compared to other active treatments. Therefore, ORs were comparable for TIRBA1%, MAL_PDT, IMQ5%_EU, IMQ3.75%, IM0.015%, CRYO, ALA_PDT, 5FU5%, 5FU4% and 5FU0.5% + SA, with overlapping credible intervals.

#### 3.4.2. Sensitivity Analysis: Single Course Data Only

Application of a single-course-of-treatment-only filter resulted in the exclusion of ALA_PDT, CRYO, IMQ3.75%, IMQ5% and MAL_PDT. Otherwise, the results of the analysis based on a subset of studies reporting complete clearance after a single course of treatment were consistent with those from the base case analysis (see Appendix A). All active treatments were associated with higher odds of complete clearance than topical placebo/vehicle (5FU0.5% + SA 6.3 [3.6–10.9]; 5FU4% 28.6 [8.9–142.7]; 5FU5% 33.1 [10.0–163.6]; DICLO3% 2.7 [1.8–4.2]; IM0.015% 13.3 [8.0–23.0]; TIRBA1% 11.1 [6.3–20.3]). 

#### 3.4.3. Sensitivity Analysis: Studies Assessing a Treatment Area of ≤25 cm^2^ Only

Application of a filter to include only studies assessing a treatment area of ≤25 cm^2^ resulted in the exclusion of ALA_PDT, IMQ3.75%, 5FU4% and 5FU5%. Results of the complete clearance analysis based on this subset of studies were broadly consistent with the base case analysis (see Appendix A). The following active treatments were associated with higher odds of complete clearance than topical placebo/vehicle: 5FU0.5% + SA 6.6 [3.4–12.9]; DICLO3% 3.1 [1.7–6.0]; IM0.015% 12.7 [7.6–22.6]; IMQ5%_EU 23.9 [10.6–55.4]; MAL_PDT 7.2 [2.4–20.9]; TIRBA1% 11.3 [5.9–23.0]. Credible intervals for CRYO were very wide and included the placebo (4.3 [1.0–18.6]. 

#### 3.4.4. Sensitivity Analysis: Single Placebo Node

A sensitivity analysis of complete clearance was conducted whereby the two placebo nodes in the base case network (PLAC_TOP and PLAC_PDT) were pooled into a single placebo node (PLAC); the results were consistent with those of the base case analysis (see Appendix A). This suggests that the equivalency of placebos is an acceptable assumption. All active treatments were associated with higher odds of complete clearance than placebo/vehicle (5FU0.5% + SA 7.6 [4.6–12.8]; 5FU4% 29.9 [9.1–134.8]; 5FU5% 34.3 [10.3–154.6]; ALA_PDT 22.1 [14.8–33.9]; CRYO 12.7 [7.0–23.6]; DICLO3% 2.8 [2.0–4.1]; IM0.015% 12.3 [8.2–19.0]; IMQ3.75% 8.5 [3.5–21.8]; IMQ5%_EU 17.4 [9.2–33.3]; MAL_PDT 10.9 [7.0–16.4]; TIRBA1% 11.1 [6.3–20.7]). 

#### 3.4.5. Sensitivity Analysis: Studies Assessing Efficacy ≥8 Weeks after Treatment Only

This sensitivity analysis of complete clearance included only studies assessing efficacy ≥8 weeks after the end of treatment: 5FU4% and 5FU5% were both excluded from this sensitivity analysis as the single study of 5FU4% or 5% relevant to the European perspective [63] assessed complete clearance at four weeks following the end of treatment. In this sensitivity analysis, the following active treatments were associated with higher odds of complete clearance than topical placebo/vehicle: 5FU0.5% + SA 6.3 [3.7–10.8]; ALA_PDT 16.8 [7.4–38.1]; CRYO 7.6 [3.0–18.9]; DICLO3% 2.3 [1.4–3.8]; IM0.015% 13.5 [8.3–22.7]; IMQ3.75% 8.5 [3.7–21.8]; MAL_PDT 9.7 [4.7–19.5]; TIRBA1% 11.0 [6.4–19.8]. IMQ5%_EU (1.5 [0.2–8.8]) had very broad credible intervals and no longer appeared significantly better than placebo, a result of the removal from the network of three out of the four studies assessing IMQ5%_EU (see Appendix A).

#### 3.4.6. Assessment of Inconsistency

An assessment of inconsistency was conducted and, in general, indirect posterior ORs were consistent with direct posterior ORs, so inconsistencies had little effect on the overall outcome of the analyses. For additional information refer to Appendix A.

## 4. Discussion

This systematic review and NMA provides a comprehensive assessment of the comparative efficacy and safety of existing treatments for AK in Europe, including the novel treatment TIRBA1%.

In the base case analysis, the active treatments demonstrated higher odds of complete clearance than topical placebo/vehicle. TIRBA1% appeared superior to DICLO3% and similarly efficacious to MAL_PDT, IMQ5%_EU, IMQ3.75%, IM0.015%, CRYO, ALA_PDT, 5FU5%, 5FU4% and 5FU0.5% + SA. The same pattern of treatment effects over placebo can be seen in the sensitivity analysis including only studies assessing after a single treatment cycle, or in the sensitivity analysis assessing a treatment area of ≤25 cm^2^, designed to more closely reflect the labeled posology for TIRBA1% [20,97].

Other recent NMAs of treatments for AK found that ALA_PDT [21,98], IMQ5% [21], 5FU0.5% + SA [21,47,48], and 5FU [47,48,99] were associated with the highest probability of achieving clearance. Interestingly, a recent NMA of long-term efficacy [22] concluded that at 12-month follow-up, 5FU5% “did not show significant long-term efficacy over placebo/vehicle for participant complete…clearance”. It is possible that 5FU efficacy varies with time, with a stronger effect sooner after treatment; this may be a confounder of the analysis. The study forming the evidence body for 5FU4% and 5FU5% [63] had, however, a high risk of bias, with incomplete reporting of blinding and no reporting of the main efficacy outcome (complete clearance) for the two placebo arms. 

In the base case analysis of complete clearance, the OR for DICLO3% was lower than the other active comparators, with credible intervals overlapping with IMQ3.75% only. One factor which may have contributed to this result is the hyaluronic acid gel vehicle used in some trials of DICLO3%, which may have some efficacy of its own and lead to comparatively high response in the placebo/vehicle arm. Thus, the relative efficacy of DICLO3% may be underestimated for this reason. 

In the sensitivity analysis including only studies assessing a treatment area of ≤25 cm^2^, all active treatments showed consistently overlapping credible intervals with TIRBA1% (except DICLO3%, which showed a very small overlap in one sensitivity analysis only). The consistency of these results suggests that the comparative efficacy of treatments may be independent of the size of the treatment field. We note that, since conducting these analyses, IM0.015% has been withdrawn from use in the EU and US [26,100].

The efficacy outcome lesion count reduction was not reported in sufficient detail to allow a quantitative analysis. Any amount of treatment was associated with lesion count reductions in the intervention arms and when compared with placebo/vehicle. However, the comparisons with placebo/vehicle were sparse and the evidence was not robust (as also concluded by Steeb et al. 2021, who stated that “The mean reduction of lesions and occurrence of adverse events was poorly reported” [98]).

Like Steeb et al. 2021 [98], the current review also found reporting of data on the relative safety profiles of the interventions assessed to be inconsistent and sparse (particularly in the case of severe LSRs). However, the safety profile of TIRBA1% showed low rates of severe LSRs (<11% for any given LSR [53,54]) while IMQ5%_EU and IMQ3.75% showed relatively high rates of some severe LSRs with 31% [61] and 25% [59] of patients experiencing severe redness and erythema, respectively. No data on the incidence of any specific severe LSR were found through this review for CRYO, IM0.015%, MAL_PDT, or 5FU (4% or 5%). Overall, the lack of reporting of severe LSRs is a major weakness of the evidence base, especially given the clinical significance of these outcomes for treatment selection, which should be addressed in future work.

Reported rates of discontinuation due to AEs were low across all interventions assessed. In both the TIRBA1% trials, 0% rates of discontinuation due to TRAEs, TEAEs, local AEs or LSRs were reported in the active (353 patients) and placebo (349 patients) arms, suggesting very high treatment adherence and tolerability. Length of treatment regimen has an impact on patient experience, and it is possible that single course treatments offering similar efficacy to multiple course treatments may have an advantage in terms of cost and patient convenience [101,102]. This offers potential advantages over other existing treatments, particularly when considering the short duration of treatment with once per day application for five days which may be of particular interest for patients with limited adherence or limited capacity for self-application, a therapeutic benefit acknowledged by the Scottish Medicines Consortium [103].

### Limitations and Assumptions

Although the feasibility assessment identified which outcomes were suitable for the NMA, some remaining limitations due to heterogeneity of the studies need to be acknowledged. Sensitivity analyses were used to assess the impact of the possible main sources of heterogeneity.

Only four of the 46 studies (9%) included in the review were considered to have an overall low risk of bias [51,52,53,54], and in many of the included studies, it was not possible to truly blind the patients and/or study personnel involved in administrating the intervention. While this was unavoidable for those trials assessing interventions requiring different methods of administration, it introduced heterogeneity in the methods across the included studies and may have impacted the relative results obtained for the treatment comparisons informed by open-label studies. 

The RCTs included in this review were characterized by substantial differences in their designs. Of the studies contributing to the analysis of complete clearance (Table 1), 16 RCTs were designed to evaluate strictly one course of treatment (independently of the recovery of the patient), while another 15 RCTs presented a design that allowed for multiple courses or sessions of treatment. Within this latter group of studies, some designs were more comparable to a real-life situation. Patients were assessed by the clinician after a course of treatment and could be prescribed additional courses of treatment, as deemed appropriate for the individual patient. This design introduced a high variability in the scheduling of the drug administration both within and between studies. Previous NMAs of AK on the face and scalp also incorporated all the courses of treatment in the main analysis but acknowledged that heterogeneity was introduced with this approach [21]. We assessed the effects of this with a sensitivity analysis (Section 3.4.2 and Appendix A) including only RCTs that evaluated one course of treatment, the results of which reflected those of the base case analysis. 

Across the 35 trials relevant to a European perspective, there were 29 placebo/vehicle arms. These placebo/vehicle arms were different in their formulations (as creams, gels, ointments, patches, and placebo PDT were used) and schedules of administration. We retained topical placebo/vehicle as a separate node to placebo PDT (with cream or patch sensitizer) but assumed equivalency of all topical placebos/vehicles. This assumption was consistent with previous NMAs of treatment for AK of the face or scalp [21,22,47,48]. Another earlier NMA [48], based on a Cochrane review, assumed equivalence of all topical placebos/vehicles and placebo PDT. In order to assess the effect of assuming placebo equivalence, we performed a sensitivity analysis (see Section 3.4.4 and Appendix A). Results suggested that this assumption was reasonable and should not represent a major limitation. 

One potential source of heterogeneity in NMAs in this clinical area is the variety of timepoints at which outcomes are reported. To mitigate this potential risk, clinical input and regulatory labels were used to define a set of common timepoints of outcome reporting for each treatment. Sensitivity analysis including only studies assessing efficacy ≥8 weeks after end of treatment (Section 3.4.5) excluded studies that could be confounded by the ongoing regeneration cycle of the epidermis and confirmed that the results of the base case are robust to this source of heterogeneity for all interventions except for IMQ5%_EU (5FU4% and 5FU5% were excluded by the analysis having an earlier timepoint of outcome assessment).

Safety outcomes, including the incidence of severe LSRs, were generally inconsistently reported and it is difficult to draw conclusions from the available data regarding the relative safety profiles of the AK treatments assessed by this review. Incidence of severe LSRs is likely to have an impact on patient tolerability and treatment satisfaction, and better reporting of the incidence of severe LSRs in future studies of treatments for AK would allow more accurate conclusions to be drawn. 

Mechanical pretreatments, such as curettage, are hard to assess in an NMA because of poor reporting but should be kept in mind as a possible source of clinical heterogeneity. In trials allowing prior mild curettage this may confound assessment of the efficacy of interventions in both active and placebo/vehicle arms. As suggested by Vegter and Tolley [21], “curettage may have caused an underestimation of the true effectiveness of the active treatments in these studies”. 

All the above-mentioned factors contributed to heterogeneity in the sample. In the quantitative assessment of between-study heterogeneity, it was found to be generally low, but was still a potential concern in the evidence that informed some treatment comparisons. Random effects models were used to account for these differences between studies, and weakly informative prior distributions were used to estimate the between-study variance. The consistency of results between the base case and sensitivity analyses suggests that the findings obtained are robust to the heterogeneity present in our sample.

## 5. Conclusions

All active treatments commonly used in Europe for the treatment of AK demonstrated higher odds of complete clearance than topical placebo/vehicle, though credible intervals were generally wide and were overlapping for comparisons across most treatments. We identified some concerns on the robustness of the evidence for some interventions, such as 5FU5% and 4%. The results should be interpreted with caution considering the wide credible intervals due to limited availability of data in the networks. In the qualitative assessment of safety outcomes, TIRBA1% showed low rates of severe LSRs while IMQ5%_EU and IMQ3.75% showed relatively high rates of some severe LSRs. No data on the incidence of severe LSRs were found for CRYO, IM0.015%, MAL_PDT, or 5FU (4% or 5%). TIRBA1% showed 0% rates of discontinuation due to TRAEs, TEAEs, local AEs or LSRs across all patients treated in both included studies, indicating very high treatment adherence.

Tirbanibulin 1% ointment provides clinicians with a novel field-directed treatment option in management of patients with AK of the face and scalp, with a single short (5 day) treatment period, once daily application, good safety profile and efficacy comparable with existing topical treatments.

## Figures and Tables

**Figure 1 jcm-11-01654-f001:**
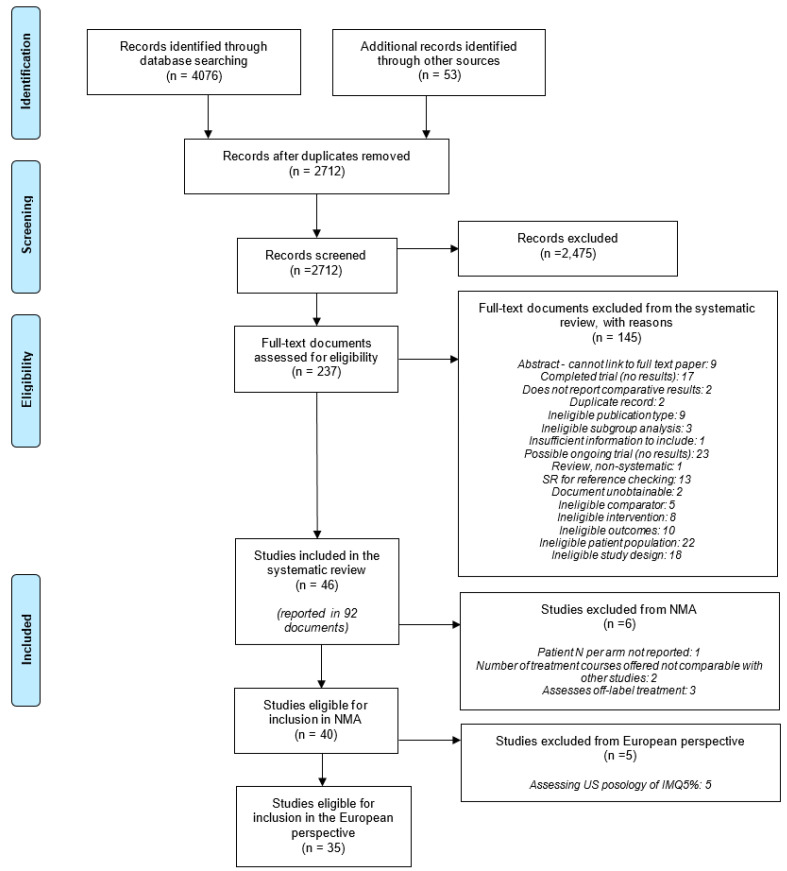
PRISMA Flow Diagram.

**Figure 2 jcm-11-01654-f002:**
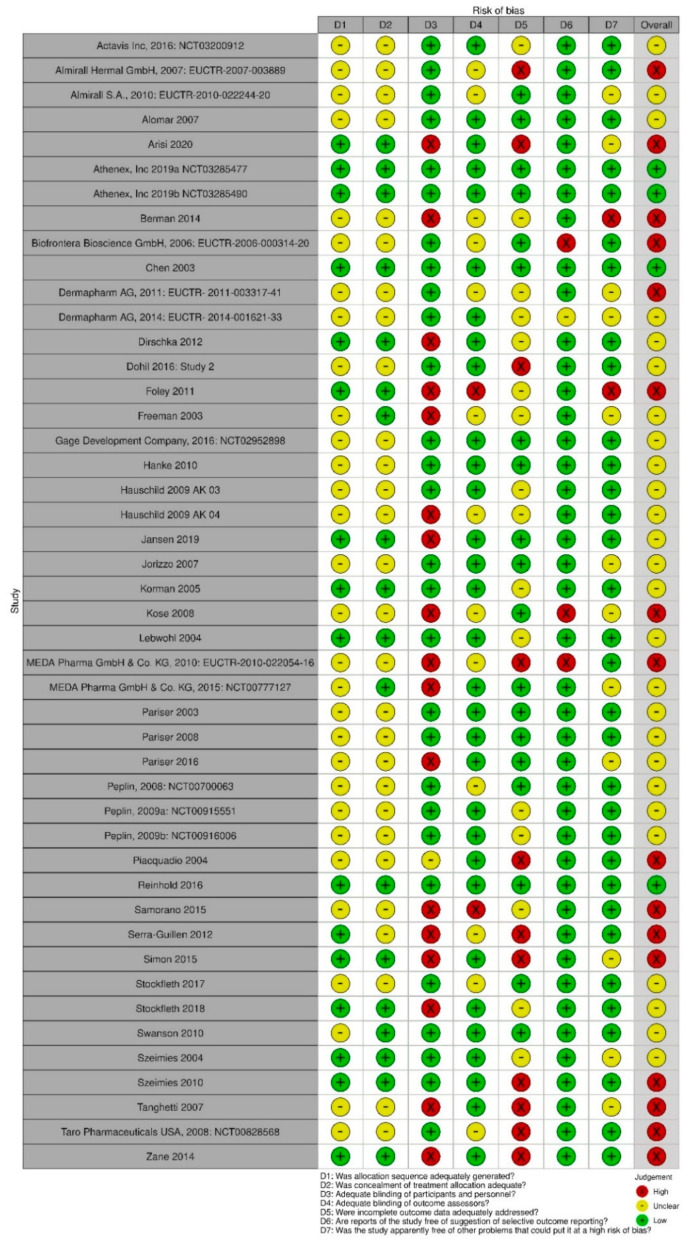
Summary Risk of Bias Assessment for the 46 studies included in the systematic review [95].

**Figure 3 jcm-11-01654-f003:**
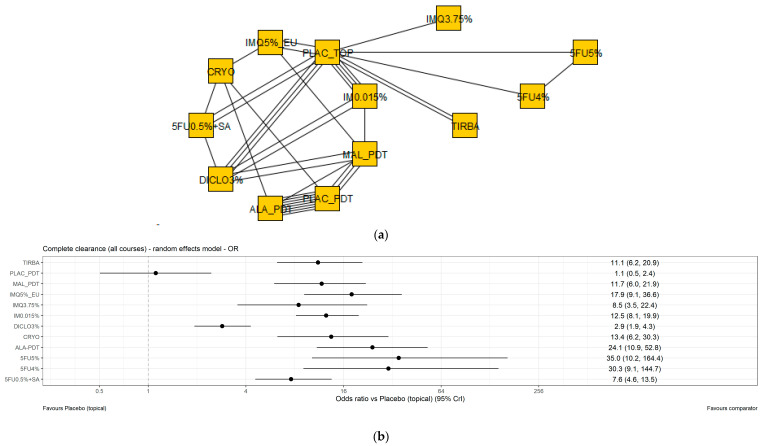
Complete clearance, European base case analysis—network diagram and forest plot. (**a**) Network diagram. (**b**) Forest plot of the interventions from the main network showing odds ratios against placebo with 95% credible intervals. 5FU0.5% + SA—5 fluorouracil 0.5% + salicylic acid 10%; 5FU4%—5 fluorouracil 4%; 5FU5%—5 fluorouracil 5%; ALA_PDT—Photodynamic therapy with 5-Aminolevulinic acid sensitizer; CRYO—Cryotherapy; DICLO3%—Diclofenac 3%; IM0.015%—Ingenol mebutate 0.015%; IMQ3.75%—Imiquimod 3.75%; IMQ5%_EU—Imiquimod 5% EU posology; MAL_PDT—Photodynamic therapy with methyl aminolevulinate sensitizer; PLAC_PDT—Photodynamic therapy with placebo sensitizer; PLAC_TOP—Topical placebo; TIRBA1%—Tirbanibulin 1%.

**Table 1 jcm-11-01654-t001:** Eligibility criteria for the global review.

	Inclusion Criteria	Exclusion Criteria
Population	Trials in adults with AK of the face and/or scalp	Trials in children Trials in populations with other skin conditions, e.g., basal cell carcinoma, squamous cell carcinoma and squamous cell carcinoma in situ (Bowen’s disease) Trials in organ transplant patients
Interventions	Cryosurgery PDT using illumination, including light-emitting diodes or daylight, with prior photosensitization with ALA or MAL Tirbanibulin 1% Diclofenac 3% Imiquimod 3.75% or 5% Ingenol mebutate 0.05% or 0.015% 5-fluorouracil 0.5% + salicylic acid 5-fluorouracil 4% or 5% Any formulations of the above will be eligible, including gels, creams, ointments, patches, sprays etc.	Trials of any other intervention For the European perspective only, studies assessing US posology of imiquimod were not included *
Comparators	Any of the interventions listed above compared to each other or to placebo/vehicle	Trials with any other comparators
Outcomes	Efficacy outcomes: Complete clearance Partial clearance Lesion count reduction Recurrence AEs and LSRs, by type and severity Discontinuations due to AEs or LSRs	For the European perspective only: partial clearance and recurrence were not assessed *
Study designs	RCTs Systematic reviews from the past five years (for reference checking only)	Intraindividual RCTs and other trial designs Phase I clinical trials Dosing studies Non-RCTs Retrospective studies Observational studies Case reports, case series, or case-control studies Narrative reviews and opinion pieces
Limits	Papers in languages other than English to be listed for information but no data extracted	

* Differences with the European perspective are underlined in the table above. AE—adverse event; AK–actinic keratosis; ALA—aminolevulinic acid; LSR—local skin reaction; MAL—methyl aminolevulinate; PDT–photodynamic therapy; RCT—randomized controlled trial.

**Table 2 jcm-11-01654-t002:** Discontinuations due to TRAEs, TEAEs, local AEs, or LSRs.

Study Identifier	Population	Period over Which Discontinuations Are Assessed	Definition of Adverse Event Leading to Reported Discontinuations	Intervention Code	N Analyzed	N (%) Patients Discontinuing
Alomar 2007 [61]	ITT	Up to 4 weeks after end of last treatment course	LSR	IMQ5%_EU	129	2 (1.6 *)
		Up to 4 weeks after end of last treatment course		PLAC_TOP	130	0
Almirall Hermal GmbH: EUCTR-2007-003889 [79]	Safety = ITT	During 12 weeks treatment period	Local TEAE	5FU0.5% + SA	187	7 (3.7 *)
		During 12 weeks treatment period		DICLO3%	185	9 (4.9 *)
		During 12 weeks treatment period		PLAC_TOP	98	1 (1.0 *)
Almirall S.A., 2010: EUCTR-2010-022244-20 [71]	Safety = ITT	Up to 150 days after star of treatment (60 days after end of treatment)	TEAE	DICLO3%	381	8 (2.1)
		Up to 150 days after start of treatment (60 days after end of treatment)		PLAC_TOP	127	5 (3.9)
		Up to 150 days after start of treatment (60 days after end of treatment)	Cutaneous side effect (erythema, oedema, pruritus, rash, skin exfoliation)	DICLO3%	381	47 (12.3)
		Up to 150 days after start of treatment (60 days after end of treatment)		PLAC_TOP	127	3 (2.4)
Athenex, Inc 2019a NCT03285477 [53]	ITT	Up to 57 days after start of treatment	TRAE	TIRBA	175	0
		Up to 57 days after start of treatment		PLAC_TOP	176	0
Athenex, Inc 2019b NCT03285490 [54]	ITT	Up to 57 days after start of treatment	TRAE	TIRBA	178	0
		Up to 57 days after start of treatment		PLAC_TOP	173	0
Chen 2003 [51]	PP	Up to 4 weeks after end of last treatment course	LSR	IMQ5%_EU	29	0
		Up to 4 weeks after end of last treatment course		PLAC_TOP	10	0
Freeman 2003 [77]	ITT	NR	Local AE	MAL_PDT	88	1 (1.1 *)
		NR		PLAC_PDT	23	NR
		NR		CRYO	89	NR
Hauschild 2009 AK 03 [78]	Safety	Up to 12 weeks after PDT session	TRAE	ALA_PDT	69	1 (1.4 *)
		Up to 12 weeks after PDT session		PLAC_PDT	34	NR
Hauschild 2009 AK 04 [78]	Safety	Up to 12 weeks after PDT session	TRAE	ALA_PDT	148	2 (1.4 *)
		Up to 12 weeks after PDT session		PLAC_PDT	49	NR
		Up to 12 weeks after CRYO session		CRYO	149	NR
Jansen 2019 [28]	Patients who completed AE diaries	During treatment or the 2 weeks after the end of treatment	Serious TRAE	5FU5%	135	0
		During treatment or the 2 weeks after the end of treatment		IMQ5%_EU	121	0
		During treatment or the 2 weeks after the end of treatment		MAL_PDT	117	0
		During treatment or the 2 weeks after the end of treatment		IM0.015%	140	0
Piacquadio 2004 [85]	Safety	Up to 12 weeks (4 weeks after last PDT)	TRAE	ALA_PDT	181	0
		Up to 12 weeks (4 weeks after last PDT)		PLAC_PDT	62	0
Reinhold 2016 [52]	ITT	Up to 12 weeks after last PDT session	TEAE	ALA_PDT	55	0
		Up to 12 weeks after last PDT session		PLAC_PDT	32	0
Simon 2015 [89]	Completers	Up to 8 weeks after end of treatment	LSR	5FU0.5% + SA	33	3 (9.1)
		Up to 11 weeks after last available CRYO session		CRYO	33	0
Stockfleth 2017 [62]	Safety = ITT	During 12 weeks treatment period	TEAE	5FU0.5% + SA	108	2 (1.9)
		During 12 weeks treatment period		PLAC_TOP	55	0
Stockfleth 2018 [75]	Safety	Up to 56 days after start of patient’s last treatment course	TRAE	IM0.015%	247	5 * (2)
		Up to 29 days after end of treatment		DICLO3%	234	14 * (6)
Swanson 2010 [59]	ITT	Up to 8 weeks after end of treatment	TRAE	IMQ3.75%	160	1 (0.6 *)
		Up to 8 weeks after end of treatment		PLAC_TOP	159	1 (0.6 *)

* indicates value calculated by reviewers. 5FU—5 Fluorouracil; AE—adverse events; AK—actinic keratosis; ALA PDT—PDT with aminolevulinic acid sensitizer; CRYO—cryotherapy; DICLO—diclofenac; IMQ—imiquimod; IM—ingenol mebutate; ITT—intent to treat; LSR—local skin reaction; MAL PDT—PDT with methyl aminolevulinate sensitizer; NR—not reported; PDT—photodynamic therapy; PLAC—placebo; PP—per protocol; SA—salicylic acid; TEAE—treatment-emergent adverse event; TIRBA—tirbanibulin; TOP—topical; TRAE—treatment-related adverse event.

**Table 3 jcm-11-01654-t003:** Patients experiencing severe LSRs.

					Proportion of Patients Experiencing Severe LSRs: *n* (%)							
Study Identifier	Population	Timepoint of Assessment	Intervention	N Analyzed	Redness/Erythema	Flaking/Scaling/Dryness	Erosion/Ulceration	Scabbing/Crusting	Vesicles	Swelling/Oedema	Itching/Pruritus	Weeping/Exudate
Almirall Hermal GmbH, 2007: EUCTR-2007-003889 [79]	ITT	During 12 weeks treatment period	5FU0.5% + SA	NR	NR	NR	NR	NR	NR	NR	9 (4.8)	NR
			DICLO3%	NR	NR	NR	NR	NR	NR	NR	13 (7)	NR
			PLAC_TOP	NR	NR	NR	NR	NR	NR	NR	0	NR
Alomar 2007 [61]	ITT	Up to 4 weeks after end of treatment	IMQ5%_EU	129	40 (31)	15 (11.6)	14 (10.9)	31 (24)	2 (1.6)	9 (7)	NR	6 (4.7)
			PLAC_TOP	130	0	1 (0.8)	1 (0.8)	2 (1.5)	0	0	NR	1 (0.8)
Athenex, Inc 2019a NCT03285477 [53]	ITT	Up to 57 days after start of treatment	TIRBA	175	5 (3)	11 (6)	0	2 (1)	1 (0.6)	1 (0.6)	1 (0.6)	NR
			PLAC_TOP	176	0	0	0	0	0	0	0	NR
Athenex, Inc 2019b NCT03285490 [54]	ITT	Up to 57 days after start of treatment	TIRBA	178	17 (10)	20 (11)	0	5 (3)	1 (0.6)	1 (0.6)	0	NR
			PLAC_TOP	173	0	1 (0.6) *	0	0	0	0	0	NR
Pariser 2016 [74]	ITT	Week 24, i.e., 16 weeks after second available session	ALA_PDT	44	0	0	NR	NR	NR	0	NR	NR
			PLAC_PDT	46	0	0	NR	NR	NR	0	NR	NR
Swanson 2010 [59]	ITT	Up to 8 weeks after end of treatment	IMQ3.75%	160	40 (25.2)	13 (8.2)	17 (10.7)	22 (13.8)	NR	9 (5.7)	NR	9 (5.7)
			PLAC_TOP	159	0	2 (1.3)	0	0	NR	0	NR	0

* indicates value calculated by reviewers. 5FU—5 Fluorouracil; AK—actinic keratosis; ALA PDT—PDT with aminolevulinic acid sensitizer; DICLO—diclofenac; IMQ—imiquimod; ITT—intent to treat; LSR—local skin reaction; NR—not reported.; PDT—photodynamic therapy; PLAC—placebo; SA—salicylic acid; TIRBA—tirbanibulin; TOP—topical.

**Table 4 jcm-11-01654-t004:** Complete clearance data eligible for inclusion in the European NMA.

Study Identifier	Population	Assesses an Area of ≤25 cm^2^?	Intervention Code for Networks	Treatment Regimen	Retreatment Offered by Study?	Timepoint of Assessment	N Experiencing Event/N Analyzed (%)
Actavis Inc, 2016: NCT03200912 [69]	PP	Yes	IM0.015%	Once daily for 3 days	No	Day 57	44/144 (30.6%)
			PLAC_TOP	Once daily for 3 days	No	Day 57	7/139 (5%)
Almirall Hermal GmbH, 2007: EUCTR-2007-003889 [79]	FAS	Yes	5FU0.5% + SA	Once daily for 12 weeks **	No	8 weeks after end of treatment	98 */177 (55.4%)
			DICLO3%	Twice daily for 12 weeks **	No	8 weeks after end of treatment/week 20	59 */183 (32%)
			PLAC_TOP	Once daily for 12 weeks **	No	8 weeks after end of treatment/week 20	14 */96 (15.1%)
Almirall S.A., 2010: EUCTR-2010-022244-20 [71]	FAS	No: up to 75 cm^2^ total	DICLO3%	Twice daily for 90 days **	No	150 days	89/380 (23.4%)
			PLAC_TOP	Twice daily for 90 days **	No	150 days	16/127 (12.6%)
Alomar 2007 [61]	ITT	Yes	IMQ5%_EU	Once daily on three alternate days per week for 4 weeks of treatment	A second 4 week course of treatment was permitted at week 8 if complete clearance was not achieved	4 weeks after end of last treatment cycle	71/129 (55%)
			PLAC_TOP	Once daily on three alternate days per week for 4 weeks of treatment		4 weeks after end of last treatment cycle	3/130 (2.3%)
Arisi 2020 [84]	PP	Yes	MAL_PDT	One session	A second session delivered only “if needed” after 3 months	90 days after final PDT	6/26 (23.07%)
			IM0.015%	Once daily for 3 days	No	90 days after end of treatment	9/30 (30%)
			DICLO3%	Twice daily for 90 days	No	90 days after end of treatment	4/28 (14.28%)
Athenex, Inc 2019a NCT03285477 [53]	ITT	Yes	TIRBA1%	Once daily for 5 days	No	Day 57	77/175 (44%)
			PLAC_TOP	Once daily for 5 days	No	Day 57	8/176 (5%)
Athenex, Inc 2019b NCT03285490 [54]	ITT	Yes	TIRBA1%	Once daily for 5 days	No	Day 57	97/178 (54%)
			PLAC_TOP	Once daily for 5 days	No	Day 57	22/173 (13%)
Biofrontera Bioscience GmbH, 2006: EUCTR-2006-000314-20 [81]	ITT	No: up to 200 cm^2^ total	ALA_PDT	One session	No	12 weeks after first (only) PDT	7 */28 (25.9%)
			PLAC_PDT	One session	No	12 weeks after first (only) PDT	1 */27 (3.7%)
Dirschka 2012 [73]	ITT	Study assessed lesion-directed treatment: area did not have to be contiguous	ALA_PDT	One session	A second session was permitted at week 12 if AKs remained	12 weeks after last PDT	194 */248 (78.2%)
			MAL_PDT	One session		12 weeks after last PDT	159 */247 (64.2%)
			PLAC_PDT	One session		12 weeks after last PDT	13 */76 (17.1%)
Dohil 2016: Study 2 [63]	ITT	No: no set target area defined	5FU4%	Once daily for 4 weeks	No	4 weeks after end of treatment	192 */353 (54.4%)
			5FU5%	Twice daily for 4 weeks	No	4 weeks after end of treatment	202 */349 (57.9%)
			PLAC_TOP	Once daily for 4 weeks	No	4 weeks after end of treatment	3/70 (4.3%)
			PLAC_TOP	Twice daily for 4 weeks	No	4 weeks after end of treatment	NR/69
Foley 2011 [86]	PP	Study assessed lesion-directed treatment: area did not have to be contiguous	IMQ5%_EU	Three times per week for 3–4 weeks	A second course was permitted 4 weeks after end of course 1 if AKs remained	40 weeks after end of treatment	17/25 (68%)
			CRYO	One session	Additional sessions permitted at 3, 6 and 9 months post-treatment if AKs remained	Unclear. Ranges from 12 to 40 weeks post final treatment	28/31 (90.3%)
Gage Development Company, 2016: NCT02952898 [68]	mITT	No: no set target area defined	DICLO3%	Twice daily for 60 days	No	90 days	56 */218 (25.7%)
			PLAC_TOP	Twice daily for 60 days	No	90 days	21 */221 (9.5%)
Hauschild 2009 AK 03 [78]	FAS	No: set target area not defined. Up to eight 4 cm^2^ patches applied	ALA_PDT	One session	No	12 weeks after first (only) PDT	41/66 (62%)
			PLAC_PDT	One session	No	12 weeks after first (only) PDT	2/33 (6%)
Hauschild 2009 AK 04 [78]	PP	No: set target area not defined. Up to eight 4 cm^2^ patches applied	ALA_PDT	One session	No	12 weeks after first (only) PDT	86/129 (67%)
			PLAC_PDT	One session	No	12 weeks after first (only) PDT	5/43 (12%)
			CRYO	One session	No	12 weeks after first (only) session	66/126 (52%)
Jorizzo 2007 [60]	ITT	Yes	IMQ5%_EU	3 times per week for 4 weeks	A second course was offered 4 weeks after end of course 1 if AKs remained	4 weeks after first treatment	33 */123 (26.8%)
			PLAC_TOP	3 times per week for 4 weeks		4 weeks after first treatment	5 */123 (4.1%)
Pariser 2003 [64]	PP	Study assessed lesion-directed treatment: area did not have to be contiguous	MAL_PDT	Two sessions one week apart	No, but all patients received the initial 2 sessions	3 months after second PDT	32/39 (82%)
			PLAC_PDT	Two sessions one week apart		3 months after second PDT	8/38 (21%)
Pariser 2008 [72]	ITT	Study assessed lesion-directed treatment: area did not have to be contiguous	MAL_PDT	Two sessions one week apart	No	3 months after second PDT	29/49 (59.2%)
			PLAC_PDT	Two sessions one week apart	No	3 months after second PDT	7/47 (14.9%)
Pariser 2016 [74]	ITT	Study assessed lesion-directed treatment: area did not have to be contiguous	ALA_PDT	One session	A second session was offered at week 8 if AKs remained	16 weeks after second (final) PDT	12/47 (25.5%)
			PLAC_PDT	One session		16 weeks after second (final) PDT	1/46 (2.2%)
Peplin, 2008: NCT00700063 [65]	ITT	No: no set target area defined	IM0.015%	Once daily for three days	No	Day 57	16/32 (50.0 *%)
			PLAC_TOP	Once daily for three days	No	Day 57	3/33 (9.1 *%)
Peplin, 2009a: NCT00915551 [66]	ITT	Yes	IM0.015%	Once daily for three days	No	Day 57	67/142 (47.2 *%)
			PLAC_TOP	Once daily for three days	No	Day 57	7/136 (5.1 *%)
Peplin, 2009b: NCT00916006 [67]	ITT	Yes	IM0.015%	Once daily for three days	No	Day 57	50/135 (37.0 *%)
			PLAC_TOP	Once daily for three days	No	Day 57	3/134 (2.2 *%)
Piacquadio 2004 [85]	PP	No: no set target area defined	ALA_PDT	One session	A second session was offered at week 8 if AKs remained	4 weeks after second (final) PDT	109/149 (73%)
			PLAC_PDT	One session		4 weeks after second (final) PDT	4/52 (8%)
Reinhold 2016 [52]	ITT	Yes	ALA_PDT	One session	A second session was offered at week 12 if AKs remained	12 weeks after last PDT	50 */55 (91%)
			PLAC_PDT	One session		12 weeks after last PDT	7 */32 (22%)
Serra-Guillen 2012 [92]	Completers	Yes	MAL_PDT	One session	No	1 month after first (only) PDT	4/40 (10%)
			IMQ5%_EU	Three times a week on alternate nights, for 4 weeks	No	1 month after end of first (only) course	9/33 (27%)
Simon 2015 [89]	Completers	Yes	5FU0.5% + SA	Once daily for 6 weeks	No	8 weeks after end of treatment	11/33 (33.3%)
			CRYO	One session	Most patients (87.9%) received a second session 3 weeks following the first session	11 weeks after second (final) session	8/32 (25%)
Stockfleth 2017 [62]	ITT	Yes	5FU0.5% + SA	Once daily for 12 weeks	No	8 weeks after end of treatment	53 */108 (49.5%)
			PLAC_TOP	Once daily for 12 weeks	No	8 weeks after end of treatment	10 */55 (18.2%)
Stockfleth 2018 [75]	The FAS included all randomized patients	Yes	IM0.015%	Once daily for 3 days	A second course was offered 8 weeks after the first course if AKs were present	Week 8 or week 17, i.e., 56 days after start of last treatment course	136 */255 (53.3%)
			DICLO3%	Twice daily for 90 days	No	End of last treatment course, defined as week 17. So, 29 days after end of treatment	58/247 (23.5%)
Swanson 2010 [59]	ITT	No: area defined as greater than 25 cm^2^	IMQ3.75%	Daily for 2 weeks	All patients received a second course 2 weeks after the end of the first course	8 weeks after end of treatment	57 */160 (35.6%)
			PLAC_TOP	Daily for 2 weeks		8 weeks after end of treatment	10 */159 (6.3%)
Szeimies 2010 [82]	FAS	Study assessed lesion-directed treatment: area did not have to be contiguous	ALA_PDT	One session	A second session was given at week 12 if AKs remained	12 weeks after last PDT	53 */80 (66.3%)
			PLAC_PDT	One session		12 weeks after last PDT	5 */40 (12.5%)
Zane 2014 [91]	PP	Study assessed lesion-directed treatment: area did not have to be contiguous	DICLO3%	Twice daily for 90 days	No	90 days after end of treatment	27/100 (27%)
			MAL_PDT	One session	A second session was given at month 3 if AKs remained	90 days (approximately 13 weeks) after final PDT	67/98 (68%)

* indicates value calculated by reviewers. ** or until the lesions had completely cleared or ulceration of the treatment area occurred. 5FU—5 Fluorouracil; AK—actinic keratosis; ALA PDT—PDT with aminolevulinic acid sensitizer; CRYO—cryotherapy; DICLO—diclofenac; FAS—full analysis set; IM—ingenol mebutate; IMQ—imiquimod; ITT—intent to treat; mITT—modified intent to treat; MAL PDT—PDT with methyl aminolevulinate sensitizer; NR—not reported; PDT—photodynamic therapy; PLAC—placebo; PP—per protocol; SA—salicylic acid; TIRBA—tirbanibulin; TOP—topical.

## Data Availability

Data supporting the quantitative analysis of complete clearance can be found in Table 4. Data supporting the qualitative analyses of additional outcomes are described in Table 2 and Table 3, and the Appendix A.

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
