# Peer review of "Comparative Efficacy and Safety of Tirbanibulin for Actinic Keratosis of the Face and Scalp in Europe: A Systematic Review and Network Meta-Analysis of Randomized Controlled Trials"

_jcm, 2022, doi:10.3390/jcm11061654_

Round 1

Reviewer 1 Report

The modified article could be published.

Author Response

Thank you for your time in reviewing this manuscript and we are pleased that you feel it is now suitable for publication.

Reviewer 2 Report

please check the literature, eg No 11 - Journal of the European Academy of Dermatology and Venereology: JEADV

Author Response

Thank you for reviewing the manuscript and for spotting the incorrectly abbreviated journals in the reference list. We have checked the list again and resolved these issues.

Reviewer 3 Report

Thank you for allowing me to review this paper entitled "Comparative efficacy and safety of tirbanibulin for actinic keratosis of the face and scalp in Europe: A systematic review and network meta-analysis of randomised controlled trials". This study aims to demonstrate that tirbanibulin 1% has a favourable safety profile and efficacy for the treatment of AK. The authors perform a network meta-analysis using data from the two current phase 3 randomised clinical trials of tirbanibulin. The topic is intriguing, and the manuscript is well performed. The introduction section is clear, although, in my opinion, the authors should stress the extreme clinical heterogeneity of AK conditions. The methodology is well described. In particular, I appreciated the content of Table 2, showing the risk of bias assessment for the studies included in the systematic review. The results are shown clearly. The limitations are correctly listed in the discussion section. First of all, the impossibility of blinding the patients and/or study personnel involved in administrating the intervention. Furthermore, as the authors said,  it is crucial to better report the incidence of severe LSRs in future studies of AK treatments and consider mechanical pretreatments as a critical source of clinical heterogeneity. 

Author Response

Thank you for reviewing the manuscript and for all your comments including those relating to the introduction. We have added a few lines to emphasise the clinical heterogeneity of AK - see lines 46-48 and 82-86. We hope you feel that these additions are satisfactory.

(Note to editors: the addition of extra content to the introduction has resulted in renumbering of the references throughout the main file, the supplementary file, and the table files. We have attached all of the updated files.)